# Detection of Intracranial Hemorrhage from Computed Tomography Images: Diagnostic Role and Efficacy of ChatGPT-4o

**DOI:** 10.3390/diagnostics15020143

**Published:** 2025-01-09

**Authors:** Mustafa Koyun, Zeycan Kubra Cevval, Bahadir Reis, Bunyamin Ece

**Affiliations:** 1Department of Radiology, Kastamonu Training and Research Hospital, Kastamonu 37150, Turkey; zeycancevval@gmail.com; 2Department of Radiology, Kastamonu University, Kastamonu 37150, Turkey; bahadirreis@hotmail.com (B.R.); bunyaminece@kastamonu.edu.tr (B.E.)

**Keywords:** ChatGPT, artificial intelligence, intracranial hemorrhage, computed tomography, radiology

## Abstract

**Background/Objectives:** The role of artificial intelligence (AI) in radiological image analysis is rapidly evolving. This study evaluates the diagnostic performance of Chat Generative Pre-trained Transformer Omni (GPT-4 Omni) in detecting intracranial hemorrhages (ICHs) in non-contrast computed tomography (NCCT) images, along with its ability to classify hemorrhage type, stage, anatomical location, and associated findings. **Methods:** A retrospective study was conducted using 240 cases, comprising 120 ICH cases and 120 controls with normal findings. Five consecutive NCCT slices per case were selected by radiologists and analyzed by ChatGPT-4o using a standardized prompt with nine questions. Diagnostic accuracy, sensitivity, specificity, positive predictive value (PPV), and negative predictive value (NPV) were calculated by comparing the model’s results with radiologists’ assessments (the gold standard). After a two-week interval, the same dataset was re-evaluated to assess intra-observer reliability and consistency. **Results:** ChatGPT-4o achieved 100% accuracy in identifying imaging modality type. For ICH detection, the model demonstrated a diagnostic accuracy of 68.3%, sensitivity of 79.2%, specificity of 57.5%, PPV of 65.1%, and NPV of 73.4%. It correctly classified 34.0% of hemorrhage types and 7.3% of localizations. All ICH-positive cases were identified as acute phase (100%). In the second evaluation, diagnostic accuracy improved to 73.3%, with a sensitivity of 86.7% and a specificity of 60%. The Cohen’s Kappa coefficient for intra-observer agreement in ICH detection indicated moderate agreement (κ = 0.469). **Conclusions:** ChatGPT-4o shows promise in identifying imaging modalities and ICH presence but demonstrates limitations in localization and hemorrhage type classification. These findings highlight its potential for improvement through targeted training for medical applications.

## 1. Introduction

Intracranial hemorrhage (ICH) is the second leading cause of stroke cases worldwide and occurs secondary to multiple factors such as trauma, infarction, aneurysm rupture, and anticoagulant therapy [1,2]. It is estimated that 37,000 to 50,000 ICH cases occur annually in the United States [3]. ICH can be classified into different types including subdural, epidural, subarachnoid, intraparenchymal, and intraventricular hemorrhages, and may lead to severe neurological damage or death if not managed appropriately [4]. A study indicates that, despite a slight decrease in ICH incidence during the ten-year period from 2000 to 2010, there was no significant reduction in mortality rates, with the 30-day mortality rate approaching 40% [5]. Half of the deaths following ICH occur within the first 24 h, and early diagnosis and treatment improve outcomes [6]. The existence of ICH constitutes a contraindication for intravenous thrombolysis therapy in patients with stroke [7]. Therefore, the early and accurate detection of ICH is crucial for patient prognosis and treatment planning.

In the detection of ICH and the identification of the cause of hemorrhage, imaging modalities such as non-contrast computed tomography (NCCT) of the brain, magnetic resonance imaging (MRI), and cerebral angiography are utilized [8]. Among these imaging modalities, NCCT is considered the gold standard for ICH detection due to its high sensitivity and is widely used in clinical practice [9]. The primary diagnostic advantage of NCCT in the early period (0–6 h) is its ability to accurately determine the presence or absence of hemorrhage [4]. Additionally, NCCT can provide valuable information in determining the hemorrhage localization, intraventricular extension, occurrence and degree of hydrocephalus and edema, presence of midline shift, and compressive effects of the hematoma on surrounding structures [9]. Currently, computed tomography (CT) images are evaluated and reported by radiologists, requiring a specific time period for accurate interpretation. The number of images that radiologists must analyze on a daily basis has grown dramatically over time; reports suggest that they are required to assess images from a single patient in as low as 3–4 s [10]. Moreover, due to the absence of round-the-clock radiologist coverage in many hospitals, CT images are evaluated by radiology residents or clinicians, particularly during night shifts [11,12]. The brief time allocated for image interpretation and the dependency of results on interpreter experience lead to both diagnostic errors and interpretation variations in ICH diagnosis [13].

Artificial intelligence (AI) technology has made rapid progress in recent years and has started to be used in many areas of our lives. The use of AI in healthcare, particularly in radiology, is steadily increasing. AI has been demonstrated to be beneficial in multiple aspects of healthcare, such as accurately identifying radiological imaging findings, improving health screening programs, and reducing medical errors [14]. An AI component integrated into radiological imaging processes is considered to have the potential to increase work efficiency, expedite diagnostic processes, and reduce human-related errors [13]. In this context, AI technologies such as deep learning algorithms have been developed to reduce workload and time spent in radiology while enhancing diagnostic power [15,16,17]. Deep learning is a machine learning approach that utilizes convolutional neural networks for solving both simple and complex tasks [18]. Hundreds of deep learning algorithms have been developed for ICH detection across different centers, supported by varying levels of evidence [3,19,20,21,22]. These deep learning algorithms have been reported to have numerous known limitations, including the requirement for large and diverse training datasets, biases in dataset compilation, poor generalizability, overfitting, limited clinical validation studies, and inability to interpret underlying mechanisms [22]. Moreover, it poses a challenge for radiologists to select and utilize one algorithm among hundreds of deep learning algorithms. At this point, the question arises whether Chat Generative Pre-trained Transformer (ChatGPT) (OpenAI, San Francisco, CA, USA), which has become highly popular in recent years and offers advanced image/audio processing capabilities, possesses the ability to interpret radiological images and participate in the diagnostic process.

ChatGPT, one of the AI technologies, was developed by OpenAI and introduced on 30 November 2022. ChatGPT is one of the large language models (LLMs) that has been trained unsupervised on extensive text data and can subsequently make inferences about relationships between words within text [23]. The initial versions of ChatGPT, ChatGPT-3 and 3.5, possessed multiple capabilities including text authorship, cross-language translation, and question answering. ChatGPT-4o was introduced on 13 May 2024 and offered image processing and inference capabilities in addition to its previous features. The presence of ChatGPT’s image processing and inference capabilities suggests its potential to evaluate radiological images. ChatGPT is considered to have the potential to assist in evaluating radiological images, detecting and characterizing abnormalities, providing preliminary diagnoses, and even recommending appropriate follow-up examinations [24].

Several studies in the literature have employed ChatGPT to evaluate mammographic images, identify distal radius fractures from radiographs, assess COVID-19 and lung cancer through chest CT imaging, and analyze ischemic stroke utilizing diffusion-weighted imaging (DWI) [25,26,27,28]. In the literature, many AI models, particularly deep learning-based ones, have been used in ICH detection (Table 1) [3,16,20,29,30]. However, our literature review revealed no studies evaluating intracranial hemorrhage detection using ChatGPT-4o. The major contributions of our study to the literature are that it is the first research evaluating the performance of ChatGPT-4o in detecting intracranial hemorrhage in brain NCCT images, and our study demonstrates the potential of a general-purpose LLM in a specialized medical imaging field.

The aim of this study is to investigate whether ChatGPT-4o can accurately detect ICH, its type, stage, localization, and associated pathologies from brain NCCT images.

## 2. Materials and Methods

### 2.1. Study Design and Case Selection

The approval for this retrospective study was obtained from the Non-Interventional Clinical Research Ethics Committee of Erzincan Binali Yildirim University (decision no: 2024-09/05; date: 11 July 2024). Our study adhered to the Checklist for Artificial Intelligence in Medical Imaging (CLAIM) guidelines [31].

Sample size calculation was performed using G*Power software (version 3.1.9.7). Power analysis was conducted with 80% statistical power and type I error probability of α = 0.05. To detect a 90% success rate for the tested diagnostic method compared to the 100% success rate of the reference group, it was determined that a total of 132 cases, with a minimum of 66 cases per group, should be included in the study.

The brain NCCT images of 4853 patients archived in our institution’s Picture Archiving Communication Systems (PACS) database between June 2023 and June 2024 were retrospectively evaluated. Considering the inclusion and exclusion criteria detailed below, a total of 240 cases were selected, comprising 120 cases with intracranial hemorrhage and 120 cases without intracranial hemorrhage (Figure 1). Case selection was performed through consensus evaluation by two radiologists with 8 and 10 years of clinical experience, respectively. Subsequently, five brain NCCT slices were selected from each case.

Within the framework of the study protocol, a standardized prompt set consisting of nine questions was developed to be presented to ChatGPT-4o (GPT-4 Omni) (OpenAI, San Francisco, CA, USA) along with brain NCCT images. Five brain NCCT slices were selected from each case and uploaded to the model along with the designated prompt, and the resultant data were documented (Figure 2, Figure 3, Figure A1 and Figure A2). The acquired data were compared with the gold standard, which was the consensus-based conclusions of radiologists. Following a two-week interval, the brain NCCT images were uploaded to ChatGPT-4o utilizing the identical prompt. This methodological approach was implemented to assess the model’s response consistency across different temporal intervals. Furthermore, to minimize potential bias effects stemming from previous analyses, ICH and control group cases were uploaded to the model in a randomized sequence.

### 2.2. Inclusion and Exclusion Criteria

Cases with ICH (epidural, subdural, parenchymal, subarachnoid, intraventricular hemorrhage) on brain NCCT slices were included in the ICH group, while cases without ICH were included in the control group. The exclusion criteria for both ICH and control groups comprised artifacts in brain NCCT slices, patient age below 18 years, the presence of masses, foreign bodies, recent cranial surgical findings, and operative materials. Additionally, for the control group, the presence of hydrocephalus, developmental brain anomalies, and areas of ischemia–infarction were established as exclusion criteria.

### 2.3. Brain NCCT Images

Brain NCCT examinations were performed using a standardized protocol on a 16-slice spiral CT (Lightspeed 16, General Electric Medical Systems, Milwaukee, WI 53201, USA) with acquisition parameters as follows: a tube voltage of 120 kVp, a 128-detector configuration, and a slice thickness of 2 mm. All images were obtained in the axial plane with brain parenchymal window settings (window width: 120; window level: 40).

### 2.4. Image Selection and Evaluation

The image upload capability of ChatGPT-4o is dependent on various factors including the number of images, image dimensions, and the volume of accompanying text, with a current size limitation of 20 megabytes per image [32]. Therefore, it was determined that five consecutive slices should be uploaded rather than all brain NCCT images of the cases to the model. From cases with ICH, 5 consecutive slices that optimally demonstrated the hemorrhagic area were selected. For the control group, sectional images were identified at anatomical planes equivalent to those selected from ICH cases.

Image formats compatible with ChatGPT-4o include JPEG, PNG, and static GIF files [29]. Therefore, the original brain NCCT images in DICOM (Digital Imaging and Communications in Medicine) format were anonymized and converted to JPEG format using AW Volumeshare 7 software (AW 4.7 version, GE Healthcare, Milwaukee, WI, USA). Irrelevant regions outside the brain were cropped and removed from the JPEG images. The JPEG images possessed variable pixel dimensions, with minimum pixel dimensions determined to be 470 × 470 pixels. Immediately prior to uploading to ChatGPT-4o for analysis, a comprehensive orientation verification was performed by radiologists. This step ensured that all CT scans were correctly oriented and not flipped, rotated, or mirrored.

### 2.5. Prompt Selection and Testing

Prior to finalizing the prompt for ChatGPT-4o, we conducted various prompt iterations. The objective was to consistently obtain detailed and descriptive responses from ChatGPT-4o that could be valuable for diagnostic evaluation. Ultimately, the following prompt consisting of nine comprehensive questions was observed to elicit complete responses from ChatGPT-4o.

### 2.6. ChatGPT Interaction and Prompting

Following the preliminary testing phase, the prompt consisting of nine questions that we established and uploaded to ChatGPT-4o was as follows:What is the name of this radiological imaging method? Are there any hemorrhages in these images? Please answer “Yes” or “No”.What type(s) of hemorrhage(s) are present? Please answer this question with “subdural”, “epidural”, “subarachnoid”, “intraparenchymal”, or “intraventricular”. If there are multiple hemorrhages in these images, please answer this question separately for each hemorrhage.If the type of hemorrhage(s) is “subdural” or “epidural”, what is the location of the hemorrhage(s)? Please specify the location with “on the patient’s right” or “on the patient’s left” followed by “frontal”, “parietal”, “temporal”, “occipital”, “frontoparietal”, “frontotemporal”, “temporoparietal”, “temporooccipital”, “posterior fossa”, etc. If there are multiple hemorrhages in these images, please answer this question separately for each hemorrhage.If the type of hemorrhage(s) is “intraparenchymal”, what is the location of the hemorrhage(s)? Please specify the location with “on the patient’s right” or “on the patient’s left” followed by “frontal”, “parietal”, “temporal”, “occipital”, “frontoparietal”, “frontotemporal”, “temporoparietal”, “temporooccipital”, “cerebellar”, “thalamus”, “pons”, “mesencephalon”, etc.If there are multiple hemorrhages in these images, please answer this question separately for each hemorrhage.If the type of hemorrhage(s) is “subarachnoid”, what is the location of the hemorrhage(s) in relation to the patient? Please specify the location with “on the patient’s right” or “on the patient’s left” followed by “frontal”, “parietal”, “temporal”, “occipital”, “frontoparietal”, “frontotemporal”, “temporoparietal”, “temporooccipital”, “cerebellar”. If there are multiple hemorrhages in these images, please answer this question separately for each hemorrhage.If the type of hemorrhage(s) is “intraventricular”, what is the location of the hemorrhage(s) in relation to the patient? Please specify the location as “3rd ventricle”, “4th ventricle”, “right lateral ventricle”, or “left lateral ventricle”. If there are multiple hemorrhages in these images, please answer this question separately for each hemorrhage.What is the phase of the hemorrhage(s)? Please answer this question with “acute”, “subacute”, or “chronic”. If there are multiple hemorrhages in these images, please answer this question separately for each hemorrhage.Are there any additional pathological findings related to the hemorrhage(s) in these images? If so, please specify the pathology/ies as “right shift”, “left shift”, “brain edema”, “3rd ventricle compression”, “right lateral ventricle compression”, “left lateral ventricle compression”, etc.

### 2.7. Executors and Readers

Both radiologists, with 8 and 10 years of experience, respectively, actively perform brain CT imaging reporting in routine clinical practice. To ensure standardization in the evaluation process, preliminary assessments were conducted on sample cases not included in the study, during which consensus on evaluation criteria was established between radiologists. The criteria for response categorization were established as follows: a “correct” response required complete concordance between ChatGPT-4o’s answers and the radiologists’ reports; a “partially correct” response indicated agreement on main findings but with some minor detail deficiencies; and an “incorrect” response was defined by discordance in main findings or the presence of diagnostically critical errors. During the evaluation process, radiologists conducted the study using medical monitors at diagnostic imaging workstations.

### 2.8. Statistical Analysis

The Statistical Package for the Social Sciences (SPSS) for Windows version 23 software (IBM SPSS Inc., Chicago, IL, USA) was used in the statistical analysis. The Kolmogorov–Smirnov test was utilized to assess normality distribution. The Mann–Whitney U test was employed to evaluate age differences between groups, while gender differences were assessed using the Chi-square test. Numerical variables with normal distribution were presented as mean ± standard deviation (SD), and categorical variables were expressed as numbers (*n*) and percentages (%). Sensitivity, specificity, positive predictive value (PPV), negative predictive value (NPV), and diagnostic accuracy rate were calculated from ChatGPT-4o’s responses. Agreement between ChatGPT-4o’s responses given at two-week intervals (intra-rater correlation) was compared using Cohen’s Kappa coefficient. The obtained Kappa coefficients were interpreted as follows: ≤0.20 indicated “very poor agreement”, 0.21–0.40 “poor agreement”, 0.41–0.60 “moderate agreement”, 0.61–0.80 “good agreement”, and 0.81–1.00 “very good agreement”. A *p* value < 0.05 was considered statistically significant.

## 3. Results

A total of 240 cases were included in the study, comprising 120 cases in the ICH group and 120 cases in the healthy control group. The mean age of the 240 cases was 63.8 ± 19.6 years. No significant differences were observed between the hemorrhage group and healthy control group in terms of age and gender (*p* > 0.05) (Table 2).

A total of 150 hemorrhagic areas were present in 120 cases in the ICH group. Among hemorrhage types, subdural hemorrhage was the most frequent with 78 (52%) cases, followed by intraparenchymal hemorrhage with 35 (23.3%) cases, while intraventricular hemorrhage was the least common with 7 (4.7%) cases (Table 3). In the ICH group, hemorrhage was unifocal in 92 (76.7%) cases and multifocal in 28 (23.3%) cases. While 57 (47.5%) cases in the ICH group had additional pathology associated with hemorrhage, 63 (52.5%) cases had no hemorrhage-related additional pathology. Some cases with hemorrhage-related additional pathology presented with single pathology, while others had multiple pathologies. A total of 85 additional pathologies were identified in 120 cases in the ICH group, and these pathologies, in order of frequency, were as follows: cerebral edema in 44 (51.8%) cases, midline shift in 23 (27%) cases, left lateral ventricle compression in 10 (11.8%) cases, and right lateral ventricle compression in 8 (9.4%) cases (Table 3).

Of the 150 hemorrhages in the ICH group, 105 (70%) were acute, 13 (8.7%) were subacute, and 27 (18%) were chronic. Additionally, three (2%) cases presented with mixed-type hemorrhage showing both acute and chronic characteristics, while two (1.3%) cases demonstrated mixed-type hemorrhage with both acute and subacute features. All mixed-type hemorrhages were subdural in nature. When hemorrhage stages were evaluated according to hemorrhage types, acute hemorrhages constituted the majority of cases across all hemorrhage types (Table 4).

In the first round, ChatGPT-4o correctly identified the imaging modality as NCCT for all cases (100%) in both ICH and control groups. Regarding the second question about the presence of hemorrhage, ChatGPT-4o correctly identified 95 (79.2%) of the 120 cases with hemorrhage, while producing false-positive results in 51 (42.5%) of the 120 cases without hemorrhage. For hemorrhage detection, ChatGPT-4o demonstrated a sensitivity of 79.2% and a specificity of 57.5%. The PPV was calculated as 65.1%, while the NPV was 73.4% (Table 5). When the same question regarding hemorrhage presence was posed to ChatGPT-4o after a two-week interval (second round), higher values were obtained for all parameters including sensitivity, specificity, PPV, NPV, and diagnostic accuracy (Table 5).

In comparing the first and second round results of ChatGPT-4o regarding hemorrhage presence, inconsistent responses were observed in 23 (19.2%) cases in the hemorrhage group and 37 (30.8%) cases in the healthy control group. There was 75% consistency between ChatGPT-4o’s first and second round results across the total study population. Analysis conducted to assess intra-rater reliability revealed moderate agreement with a Cohen’s Kappa value of 0.469 (Table 6).

In identifying hemorrhage types, ChatGPT-4o correctly identified 51 (34%) of 150 hemorrhagic areas, partially correctly identified 7 (4.7%), and incorrectly identified 64 (42.6%). The partially correct category was used for cases where ChatGPT-4o responded “subdural + intraparenchymal hemorrhage” in cases with only subdural hemorrhage. The remaining 28 (18.7%) hemorrhages belonged to 25 false-negative cases where ChatGPT-4o indicated no hemorrhage was present. Among hemorrhage types, ChatGPT-4o demonstrated the highest accuracy rate for intraparenchymal hemorrhage (25/35, 71.4%). Conversely, subarachnoid hemorrhage showed the lowest accuracy rate (1/18, 5.5%) (Table 7).

Regarding hemorrhage stages, ChatGPT-4o responded “acute” for all 95 cases (100%) where it detected hemorrhage, regardless of whether the hemorrhage was unifocal or multifocal, and did not provide detailed specifications about which hemorrhage was designated as acute in multifocal cases. Consequently, detailed statistical data regarding hemorrhage stages according to hemorrhage locations could not be obtained.

ChatGPT-4o successfully identified only 11 (7.3%) of 150 hemorrhage locations with complete accuracy. The remaining 139 (92.7%) hemorrhage locations were not correctly identified.

Regarding hemorrhage-associated pathologies, ChatGPT-4o correctly identified 11 (47.8%) of 23 midline shifts and 29 (65.9%) of 44 cases of cerebral edema in the ICH group. However, it failed to detect both right and left lateral ventricle compressions (Table 8). Additionally, it incorrectly reported third ventricle compression in seven cases and hydrocephalus in two cases where these findings were not present.

## 4. Discussion

In this study, we evaluated the ability of ChatGPT-4o to detect hemorrhages, classify their types, stages, and localizations, as well as identify the presence of any pathologies accompanying the hemorrhages on brain NCCT images. To the best of our knowledge, this is the first study in the literature to assess intracranial hemorrhages using ChatGPT-4o.

For the question regarding the type of imaging modality, ChatGPT-4o provided correct answers in all cases without any errors. Since only brain NCCT images were used in this study, it was not possible to test whether ChatGPT-4o could identify other imaging modalities. The primary finding of our study reveals that ChatGPT-4o demonstrated notable diagnostic performance in detecting intracranial hemorrhage (ICH) on brain NCCT images. The model exhibited significant diagnostic efficacy in evaluating the presence of ICH, with a sensitivity of 79.2% and a specificity of 57.5%. However, the false positive rate of 42.5% indicates a considerable tendency of the model to generate incorrect diagnoses. In the study conducted by Daiwen et al. using the previous version of ChatGPT (ChatGPT-4), it was shown that the model could identify ICHs at a rate of 72.6%, which shows considerable similarity to our rate [29]. A review of the literature shows that deep learning-based AI models have achieved high diagnostic performance in the detection of ICH. For instance, one previous study reported that a deep learning model reached a sensitivity of 95.89% and a specificity of 95.33% for intracranial hemorrhage detection [16]. Similarly, another deep learning-based study reported sensitivity and specificity rates of 78% and 80%, respectively, for the detection of intracranial hemorrhage [30]. Additionally, FDA-cleared and CE-marked triage and notification software (AIDOC) has been reported to detect intracranial hemorrhage (ICH) with a sensitivity of 89–95% and a specificity of 94–99% [3,33]. In the deep learning-based study by Heit et al., a sensitivity of 95.6% and a specificity of 95.3% were obtained in ICH detection [20]. Compared to these deep learning-based studies, ChatGPT-4o achieved relatively higher sensitivity and specificity in hemorrhage detection, despite not having been specifically trained for this purpose, thus highlighting its potential if appropriately trained in the future. Indeed, an improvement in the diagnostic parameters of the model was observed in the second-round responses compared to the first-round ones. This improvement suggests that the model may demonstrate adaptability in its learning and evaluation processes. However, the model’s tendency to repeat phrases from previous interactions could also be a contributing factor to this observed improvement [34].

The analysis conducted to evaluate intra-rater reliability revealed that ChatGPT-4o demonstrated a moderate level of consistency between the two rounds in assessing the presence of hemorrhage (κ = 0.469). This result is a significant indicator for questioning the model’s stability, as it highlights inconsistencies in its evaluations conducted at different time intervals. Consequently, it underscores that ChatGPT is not yet a fully reliable tool for determining the presence of hemorrhage.

In the evaluation of more specific clinical parameters such as the type, localization, and stage of hemorrhage, notable variations were observed in the performance of ChatGPT-4o. The low accuracy rate in determining hemorrhage localization is particularly striking. Hemorrhage localization is critical information, especially for surgical planning, and inaccuracies in this parameter could lead to life-threatening consequences. While high accuracy rates were achieved in assessing the stage of hemorrhage, it was observed that ChatGPT-4o consistently classified all cases as acute, thereby neglecting subacute and chronic hemorrhages. In the study by Daiwen et al., a low identification rate (37.3%) was also reported for chronic subdural hemorrhages [29]. Acute hemorrhages are relatively easier to identify as they appear hyperdense compared to the brain parenchyma. However, the isodense nature of subacute hemorrhages with the brain parenchyma and chronic hemorrhages with cerebrospinal fluid likely contributed to ChatGPT-4o’s inability to recognize these stages of hemorrhage. This suggests that, while the model is more adept at understanding general concepts and processes, it still lacks the capability to accurately interpret more complex and specific medical information.

Regarding hemorrhage types, ChatGPT-4o was generally able to identify subdural, intraparenchymal, and intraventricular hemorrhages. However, it struggled to detect subarachnoid and epidural hemorrhages. The study by Daiwen et al. demonstrates that ChatGPT showed similarly high success rates in detecting intraparenchymal (86.9%) and acute subdural hemorrhages (74.4%), as in our study. However, while in our study ChatGPT’s detection rates for epidural and subarachnoid hemorrhages were quite low (1/12, 8% and 1/18, 5.5%, respectively), their study found notably higher rates (89% and 71.5%, respectively) [29]. Similar to our study, in a deep learning-based study, intraparenchymal hemorrhages were identified with higher accuracy, whereas epidural hemorrhages could not be detected as effectively [3]. In another deep learning-based study, subarachnoid hemorrhages were similarly detected with lower accuracy, as observed in our study [35]. In light of these findings, we conclude that intraparenchymal hemorrhages are more easily recognized due to the contrast difference they create with brain tissue. However, the challenges in detecting subarachnoid hemorrhages are suggested to be associated with beam-hardening artifacts and partial-volume artifacts [35]. We believe that the failure of ChatGPT-4o in detecting epidural and subarachnoid hemorrhages could serve as a separate subject for further research.

Machine learning and deep learning AI applications have achieved considerable success in diagnosing from medical images. In the meta-analysis by Jung et al., the overall pooled sensitivity and specificity for bone fractures using these AI applications were found to be 91% (95% CI: 88%, 93%) and 90% (95% CI: 88%, 92%), respectively [36]. Furthermore, these AI applications can perform quantitative analysis in the diagnosis of interstitial lung diseases [37], detect brain hemorrhages with over 90% accuracy [33,35], and identify brain metastases from magnetic resonance images with up to 89% accuracy [38], and have found many clinical applications in medical imaging beyond these, demonstrating remarkable success in these areas [39,40].

However, deep learning-based AI models used in pathology detection from radiological images are purpose-specific models. They are optimized only for a particular clinical problem or imaging type. In contrast, ChatGPT-4o, as a general-purpose artificial intelligence model, has the potential to perform versatile tasks.

While developing and implementing purpose-specific AI models requires high costs and advanced technical infrastructure, general-purpose models like ChatGPT-4o offer a more accessible and cost-effective alternative.

There are dozens of purpose-specific AI models developed for ICH detection in the literature, and this diversity may cause users to experience difficulty in selecting the right model [16,19,35]. Therefore, we think that popular and general-purpose AI models like ChatGPT-4o have a higher potential to be preferred by users.

Our study shows, in light of the data we obtained, that ChatGPT-4o still has some significant limitations for clinical use. The model cannot demonstrate sufficient sensitivity and precision in detecting certain hemorrhage types like subarachnoid hemorrhage and certain hemorrhage stages like chronic phase. Another limitation of the model is its lack of ability to evaluate sequential image series while analyzing CT images. Additionally, the model requires an appropriate prompt before use. If the given prompt is not suitable, the model’s diagnostic performance may be affected. Finally, the use of ChatGPT-4o in ICH detection may bring ethical and legal debates that might not be universally accepted.

This study contributes to the theoretical knowledge base in AI and radiology by demonstrating the potential of LLMs like ChatGPT in medical image analysis. ChatGPT-4o’s success in detecting ICH from NCCT images shows significant potential for the future evaluation of radiological images and the reduction in radiologists’ workload. Particularly in healthcare centers experiencing radiologist shortages, personnel with limited clinical experience can benefit from LLMs like ChatGPT in the initial assessment of brain NCCT images. Young radiologists in the process of gaining experience can use this technology as a supportive tool to avoid missing important findings. In this way, workflow can become more efficient and patient care quality may improve. The results of our study will guide the development of similar AI systems and their integration into clinical practice.

Our study has certain limitations. The first is that, due to ChatGPT-4o’s limited capacity to process images, the model was unable to evaluate all brain NCCT images from each case. If the complete set of brain NCCT images were uploaded, the diagnostic accuracy might improve. Another limitation is that the original DICOM images had to be converted to JPEG format for analysis by ChatGPT. While this conversion was a necessary step for uploading the images to the model, we believe it might have compromised image quality, potentially affecting the model’s performance and leading to misinterpretations [27]. Additionally, the selection process of brain CT images, which were determined by radiologists and subjected to specific exclusion criteria, may have introduced selection bias. Furthermore, excluding cases with conditions that could mimic hemorrhage on brain NCCT images, such as masses, metallic clips, and foreign bodies, can be considered another limitation of the study [41]. Excluding these cases from the study may have impacted the model’s sensitivity and specificity values. The size of the hemorrhage is crucial for treatment planning. However, as the model lacks a tool for measuring hemorrhage size, no data could be obtained in this regard.

## 5. Conclusions

In conclusion, ChatGPT-4o has been evaluated as a promising auxiliary tool for detecting ICH on brain NCCT images. Considering the current limitations of the model, it should be used cautiously in clinical decision-making processes. Continuous training and development through future studies are essential to improve the model’s accuracy, sensitivity, and specificity, as well as to minimize inconsistencies. In clinical practice, the results generated by ChatGPT-4o should be assessed in collaboration with radiologists, and care must be taken in making the final decision.

## Figures and Tables

**Figure 1 diagnostics-15-00143-f001:**
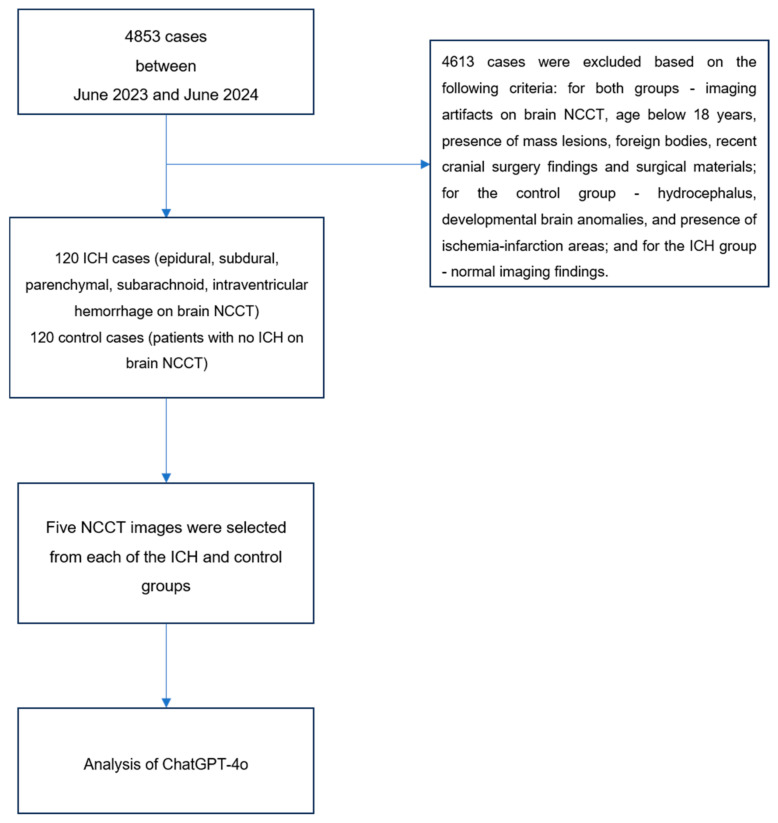
Flowchart of patient selection and ChatGPT-4o upload process.

**Figure 2 diagnostics-15-00143-f002:**
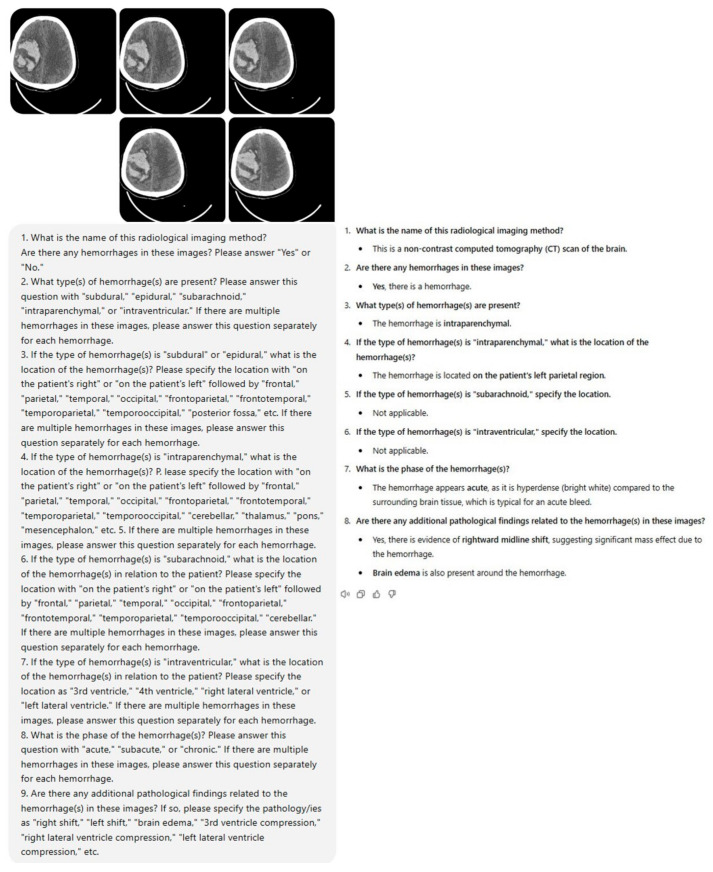
The evaluation of a case from the ICH group by ChatGPT-4o.

**Figure 3 diagnostics-15-00143-f003:**
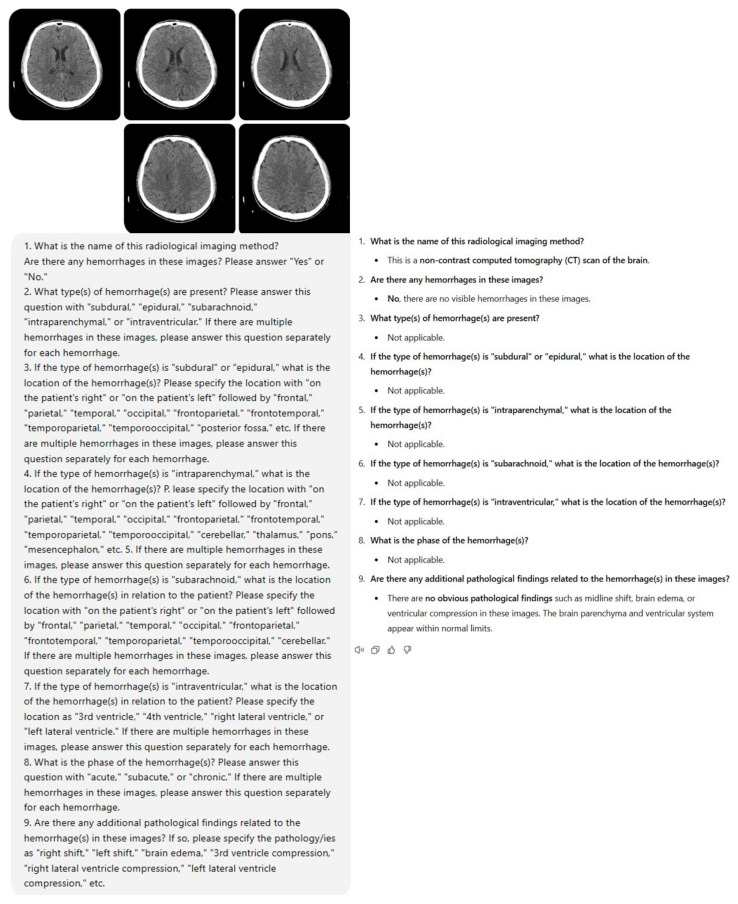
The evaluation of a case from the healthy control group by ChatGPT-4o.

**Table 1 diagnostics-15-00143-t001:** Summary table showing diagnostic performance of AI models used in ICH detection.

Author and Year	AI Model Used	Study Objective	Sample Size	Main Findings
Ginat et al., 2020 [3]	Deep learning	ICH detection from NCCT images	373 cases	ICH detection Sensitivity: 88.7%, Specificity: 94.2%
Yun et al., 2023 [16]	Deep learning	ICH detection from NCCT images	49,841 cases	ICH detection Sensitivity: 95.9%, Specificity: 95.3%
Heit et al., 2024 [20]	Deep learning	ICH detection from NCCT images	308 cases	ICH detection Sensitivity: 95.6%, Specificity: 95.3%
Daiwen et al., 2024 [29]	ChatGPT-4	ICH detection from NCCT images	208 cases	72.6% ICH detection rate
Arbabshirani et al., 2018 [30]	Deep learning	ICH detection from NCCT images	46,583 cases	ICH detection Sensitivity: 78%, Specificity: 80%

ICH: intracranial hemorrhage; AI: artificial intelligence; NCCT: non-contrast computed tomography.

**Table 2 diagnostics-15-00143-t002:** Demographic data of ICH group and healthy control group cases.

	ICH Group (*n* = 120)	Healthy Control Group (*n* = 120)	*p* Value
Gender *, *n* (%)			
Female	39 (32.5)	40 (33.3)	0.891
Male	81 (67.5)	80 (66.7)	
Age **, years, Mean ± SD	63.95 ± 19.58	63.73 ± 18.81	0.877

* Ki-kare Test; ** Mann–Whitney U test; SD: standard deviation; ICH: intracranial hemorrhage.

**Table 3 diagnostics-15-00143-t003:** Distribution of hemorrhage types, locations, and associated pathologies in ICH group.

	*n* (%)
Hemorrhage type (hemorrhagic areas, *n* = 150)	
Subdural	78 (52)
Epidural	12 (8)
Intraventricular	7 (4.7)
Intraparenchymal	35 (23.3)
Subarachnoid	18 (12)
Hemorrhage location (cases, *n* = 120)	
Unifocal	92 (76.7)
Multifocal	28 (23.3)
Associated pathologies (*n* = 85)	
Cerebral edema	44 (51.8)
Midline shift	23 (27)
Right lateral ventricle compression	8 (9.4)
Left lateral ventricle compression	10 (11.8)

**Table 4 diagnostics-15-00143-t004:** Distribution of hemorrhage stages according to hemorrhage types in ICH group.

Hemorrhage Stage	Subdural *n* (%)	Epidural *n* (%)	Intraventricular *n* (%)	Intraparenchymal *n* (%)	Subarachnoid *n* (%)	Total *n* (%)
Acute	38 (25.3)	10 (6.7)	7 (4.7)	32 (21.3)	18 (12)	105 (70)
Subacute	11 (7.3)	1 (0.7)	0 (0)	1 (0.7)	0 (0)	13 (8.7)
Chronic	24 (16)	1 (0.7)	0 (0)	2 (1.3)	0 (0)	27 (18)
Acute–subacute	2 (1.3)	0 (0)	0 (0)	0 (0)	0 (0)	2 (1.3)
Acute–chronic	3 (2)	0 (0)	0 (0)	0 (0)	0 (0)	3 (2)
Total	78 (52)	12 (8)	7 (4.7)	35 (23.3)	18 (12)	150 (100)

**Table 5 diagnostics-15-00143-t005:** First and second round results of ChatGPT-4o in evaluation of hemorrhage presence.

	True Positive, *n* (%)	False Positive, *n* (%)	True Negative, *n* (%)	False Negative, *n* (%)	Sensitivity, %	Specificity, %	PPV, %	NPV, %	Diagnostic Accuracy, %
ChatGPT-4o, Round 1	95 (79.2)	51 (42.5)	69 (57.5)	25 (20.8)	79.2	57.5	65.1	73.4	68.3
ChatGPT-4o, Round 2	104 (86.7)	48 (40)	72 (60)	16 (13.3)	86.7	60.0	68.4	81.8	73.3

PPV: positive predictive value; NPV: negative predictive value.

**Table 6 diagnostics-15-00143-t006:** Comparison of first and second round ChatGPT-4o results in evaluation of hemorrhage presence.

Hemorrhage Presence Assessment							
ICH Group (*n* = 120)				Healthy Control Group (*n* = 120)			
		ChatGPT-4o, Round 1				ChatGPT-4o, Round 1	
		Negative	Positive			Negative	Positive
ChatGPT-4o, Round 2	Negative	9	7	ChatGPT-4o, Round 2	Negative	52	20
	Positive	16	88		Positive	17	31

Cohen’s Kappa value: 0.469.

**Table 7 diagnostics-15-00143-t007:** Results of ChatGPT-4o in detection of hemorrhage types.

Hemorrhage Type (*n* = 150)	Correct, *n* (%)	Partially Correct, *n* (%)	Incorrect, *n* (%)	False Negative Cases, *n* (%)	Total, *n* (%)
Subdural	21 (14)	7 (4.7)	26 (17.3)	24 (16)	78 (52)
Epidural	1 (0.7)	0 (0)	10 (6.6)	1 (0.7)	12 (8)
Intraventricular	3 (2)	0 (0)	4 (2.7)	0 (0)	7 (4.7)
Intraparenchymal	25 (16.6)	0 (0)	9 (6)	1 (0.7)	35 (23.3)
Subarachnoid	1 (0.7)	0 (0)	15 (10)	2 (1.3)	18 (12)
Total	51 (34)	7 (4.7)	64 (42.6)	28 (18.7)	150 (100)

**Table 8 diagnostics-15-00143-t008:** ChatGPT-4o results regarding pathologies associated with hemorrhage.

Associated Pathologies	True Positive, *n* (%)	False Negative, *n* (%)
Midline shift, *n* = 23	11 (47.8)	12 (52.2)
Cerebral edema, *n* = 44	29 (65.9)	15 (34.1)
Right lateral ventricle compression, *n* = 8	0 (0)	8 (100)
Left lateral ventricle compression, *n* = 10	0 (0)	10 (100)

## Data Availability

The data presented in this study are available on request from the corresponding author. The data are not publicly available due to privacy.

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
