# Peer review of "Detection of Intracranial Hemorrhage from Computed Tomography Images: Diagnostic Role and Efficacy of ChatGPT-4o"

_diagnostics, 2025, doi:10.3390/diagnostics15020143_

Round 1
Reviewer 1 Report
Comments and Suggestions for Authors
Kudos to the authors for their ambitious effort to test a general-purpose AI model in a specialized medical domain. I understand that it involves a lot of manual effort and diligence.
While there is interest among AI enthusiasts, very few put in the effort to test this out in a structured manner, in that sense, this is a strong study.
However, there are several methodological flaws in the study. My biggest concern is the 'bright pixel' problem. The model's relatively higher sensitivity for detecting hemorrhages, especially acute ones, might stem from a spurious correlation between brighter regions on NCCT and the presence of hemorrhage. This "bright pixel problem" suggests that the model could be over-relying on pixel intensity as a proxy for hemorrhage rather than understanding the broader anatomical or pathological context. This bias is further evidenced by the model’s inability to reliably classify subdural, subarachnoid, and epidural haemorrhages. The authors should include few cases with non-hemorrhagic bright areas (e.g., calcifications, dense bone, metallic artifacts) to test the model’s ability to differentiate true hemorrhage from confounding features. Add contrast-enhanced studies to differentiate hemorrhagic and non-hemorrhagic bright regions.
The paper also lacks a deeper comparison to existing AI models for ICH detection. It also does not critically assess the novelty of using ChatGPT-4o compared to purpose-built radiology AI solutions. Including more references to benchmark studies would strengthen the context. The conclusions overstate the model's potential for clinical use.
Author Response
Dear Reviewer,
We sincerely thank you for thoroughly reviewing our study and providing constructive feedback. We have carefully evaluated all your comments and suggestions and revised the manuscript accordingly.
All changes made in the revised manuscript have been marked using the track changes feature, allowing you to easily follow the corrections. Below, you can find our detailed responses to each of your suggestions and the corresponding changes we made in the manuscript listed as bullet points.
For some of your suggestions, there were points that we could not fully implement due to time constraints and methodological limitations of our study. We have explained these situations with their justifications and noted them as valuable suggestions for future studies.
We thank you again for your contributions to the review process and your constructive approach. If you think any additional clarification or correction is needed, please do not hesitate to let us know.
Best regards.
Comment 1.
Kudos to the authors for their ambitious effort to test a general-purpose AI model in a specialized medical domain. I understand that it involves a lot of manual effort and diligence.
While there is interest among AI enthusiasts, very few put in the effort to test this out in a structured manner, in that sense, this is a strong study.
However, there are several methodological flaws in the study. My biggest concern is the 'bright pixel' problem. The model's relatively higher sensitivity for detecting hemorrhages, especially acute ones, might stem from a spurious correlation between brighter regions on NCCT and the presence of hemorrhage. This "bright pixel problem" suggests that the model could be over-relying on pixel intensity as a proxy for hemorrhage rather than understanding the broader anatomical or pathological context. This bias is further evidenced by the model’s inability to reliably classify subdural, subarachnoid, and epidural haemorrhages. The authors should include few cases with non-hemorrhagic bright areas (e.g., calcifications, dense bone, metallic artifacts) to test the model’s ability to differentiate true hemorrhage from confounding features. Add contrast-enhanced studies to differentiate hemorrhagic and non-hemorrhagic bright regions.
Response 1:
We understand your concern regarding the 'bright pixel issue' and acknowledge its significance. However our main goal was to conduct pioneering research in an area not yet explored in literature and was to evaluate the performance of ChatGPT-4o, a general-purpose large language model, in detecting intracerebral hemorrhage. While designing our study, we specifically excluded cases with hyperdense structures (such as calcification, metallic foreign bodies) that could mimic hemorrhage in NCCT images. Our main purpose in doing this was to better test whether a model that has not been investigated before can make a diagnosis by focusing only on hemorrhage. For this reason, we specifically excluded mimickers from the study by setting them as an exclusion criterion. Your suggestion to include cases with calcifications, dense bone structures, and metallic artifacts is valuable and will guide future studies. However the potential impact of conditions that could be confused with hemorrhage on ChatGPT-4o's sensitivity and specificity parameters has been addressed in our study's limitations section. For the same reason as the other mimickers above, we designed the study especially on non-contrast CT images. Again we think that this is a subject that can be the subject of further studies based on your suggestions.
The relevant sentence in the limitations section of our study is as follows:
- Furthermore, excluding cases with conditions that could mimic hemorrhage on brain NCCT images, such as masses, metallic clips, and foreign bodies, can be considered an-other limitation of the study [36]. Excluding these cases from the study may have im-pacted the model's sensitivity and specificity values.
Comment 2.
The paper also lacks a deeper comparison to existing AI models for ICH detection. Including more references to benchmark studies would strengthen the context.
Response 2:
We acknowledge your criticism regarding the lack of a deeper comparison of existing AI models for ICH detection and your suggestions to add more references to comparative studies. In this context, we would like to note that we have made more comparisons with other studies in the literature in the discussion section of our study and added more references to the discussion section. The new sentences we added to the discussion section are as follows.
-In the study conducted by Daiwen et al. using the previous version of ChatGPT (ChatGPT-4), it was shown that the model could identify ICHs at a rate of 72.6%, which shows considerable similarity to our rate [29].
-In the deep learning-based study by Heit et al., sensitivity of 95.6% and specificity of 95.3% were obtained in ICH detection [20].
-In the study by Daiwen et al., a low identification rate (37.3%) was also reported for chronic subdural hemorrhages [29].
-The study by Daiwen et al. demonstrates that ChatGPT showed similarly high success rates in detecting intraparenchymal (86.9%) and acute subdural hemorrhages (74.4%), as in our study. However, while in our study ChatGPT's detection rates for epidural and subarachnoid hemorrhages were quite low (1/12, 8% and 1/18, 5.5% respectively), their study found notably higher rates (89% and 71.5% respectively) [29].
Comment 3.
It also does not critically assess the novelty of using ChatGPT-4o compared to purpose-built radiology AI solutions.
Response 3:
We agree with your criticism that the paper does not critically evaluate the novelty of using ChatGPT-4o compared to purpose-built AI solutions, and we would like to note that we have made the following additions to the discussion section of our manuscript.
-Deep learning-based AI models used in pathology detection from radiological images are purpose-specific models. They are optimized only for a particular clinical problem or imaging type. In contrast, ChatGPT-4o, as a general-purpose artificial intelligence model, has the potential to perform versatile tasks.
-While developing and implementing purpose-specific AI models requires high costs and advanced technical infrastructure, general-purpose models like ChatGPT-4o offer a more accessible and cost-effective alternative.
- There are dozens of purpose-specific AI models developed for ICH detection in the literature, and this diversity may cause users to experience difficulty in selecting the right model [16,19,35]. However, we assess that popular and general-purpose AI models like ChatGPT-4o have a higher potential to be preferred by users.
Comment 4.
The conclusions overstate the model's potential for clinical use.
Response 4:
Regarding your criticism that the results overstate the model's potential for clinical use: As stated in our paper, this study is a starting point and does not claim the model's readiness for clinical use. Rather, it aims to establish groundwork for future research and demonstrate potential in this field. Future studies evaluating more complex cases and confounding factors, as you suggested, will help better understand the model's clinical potential. To address this criticism, we have added the following paragraph about clinical use to our discussion section.
- Our study shows, in light of the data we obtained, that ChatGPT-4o still has some significant limitations for clinical use. The model cannot demonstrate sufficient sensitivity and precision in detecting certain hemorrhage types like subarachnoid hemorrhage and certain hemorrhage stages like chronic phase. Another limitation of the model is its lack of ability to evaluate sequential image series while analyzing CT images. Additionally, the model requires an appropriate prompt before use. If the given prompt is not suitable, the model's diagnostic performance may be affected. Finally, the use of ChatGPT-4o in ICH detection may bring ethical and legal debates that might not be universally accepted.
We are grateful for the time you have dedicated to our study. Your recommendations have guided us in preparing a more enhanced and structured version of our study. We hope that our revisions meet your expectations.
Sincerely

Reviewer 2 Report
Comments and Suggestions for Authors
Dear Authors,
Thank you for submitting the article, After reviewing the article I am going to share the following comments to improve the quality of the paper.
1. The introduction section needs to be enhanced as well as literature review with a summary table.
2. Add major contributions as points at the end of the introduction section.
3. Add theoretical and practical implications of this study.
4. Better to add some more samples like Figure 2 and Figure 3 but in the appendix section of the paper.
5. What about considering the high quality images (Recommended but not necessary: the quality of the image can be enhanced (different Weights are available on HuggingFace) before passing it to the ChatGPT)?
6. What about to consider the latest version of ChatGPT (OpenAI's o1)?
7. It is recommended but not necessary that it's better to consider some ChatGPT comparison tools (Google Gemini:) also for this study
Author Response
Dear Reviewer,
We sincerely thank you for thoroughly reviewing our study and providing constructive feedback. We have carefully evaluated all your comments and suggestions and revised the manuscript accordingly.
All changes made in the revised manuscript have been marked using the track changes feature, allowing you to easily follow the corrections. Below, you can find our detailed responses to each of your suggestions and the corresponding changes we made in the manuscript listed as bullet points.
For some of your suggestions, there were points that we could not fully implement due to time constraints and methodological limitations of our study. We have explained these situations with their justifications and noted them as valuable suggestions for future studies.
We thank you again for your contributions to the review process and your constructive approach. If you think any additional clarification or correction is needed, please do not hesitate to let us know.
Best regards.
Comment 1. The introduction section needs to be enhanced as well as literature review with a summary table.
Response 1.
We have addressed your suggestion by enhancing the introduction section with additional literature data and including a summary table as requested. The new sentences we added to the introduction section are as follows.
- In the literature, many AI models, particularly deep learning-based ones, have been used in ICH detection [3,16,20,29,30] (Table 1).
Comment 2. Add major contributions as points at the end of the introduction section.
Response 2:
Based on your suggestions, we have made the following additions to the introduction section of our manuscript:
-The major contributions of our study to the literature are that it is the first research evaluating the performance of ChatGPT-4o in detecting intracranial hemorrhage in brain NCCT images and our study demonstrates the potential of a general-purpose LLM in a specialized medical imaging field.
Comment 3. Add theoretical and practical implications of this study.
Response 3:
Based on your suggestions, we have made the following additions to the discussion section of our manuscript:
-This study contributes to the theoretical knowledge base in AI and radiology by demonstrating the potential of LLMs like ChatGPT in medical image analysis. ChatGPT-4o's success in detecting ICH from NCCT images shows significant potential for future evaluation of radiological images and reducing radiologists' workload. Particularly in healthcare centers experiencing radiologist shortages, personnel with limited clinical experience can benefit from LLMs like ChatGPT in the initial assessment of brain NCCT images. Young radiologists in the process of gaining experience can use this technology as a supportive tool to avoid missing important findings. This way, workflow can become more efficient and patient care quality may improve. The results of our study will guide the development of similar AI systems and their integration into clinical practice.
Comment 4. Better to add some more samples like Figure 2 and Figure 3 but in the appendix section of the paper.
Response 4:
Based your suggestion, we have added two supplementary figures to the appendix section of the manuscript to better visualize our findings and provide readers with more comprehensive examples. These supplementary figures:
1. Appendix A1 (Figure A1): Evaluation of a case with right parietal chronic subdural hematoma by ChatGPT-4o.
2. Appendix A2 (Figure A2): Evaluation of a case with right frontoparietal acute subdural hematoma by ChatGPT-4o.
These supplementary figures better demonstrate ChatGPT-4o's ability to detect and evaluate different ICH types. The examples show how the model interprets hemorrhages in various locations and stages. Your suggestion has enriched our manuscript's visual content and improved the understanding of our findings.
Figure A1: Evaluation of a case with right parietal chronic subdural hematoma by ChatGPT-4o.
Figure A2: Evaluation of a case with right frontoparietal acute subdural hematoma by ChatGPT-4o.
Comment 5. What about considering the high quality images (Recommended but not necessary: the quality of the image can be enhanced (different Weights are available on HuggingFace) before passing it to the ChatGPT)?
Response 5:
Using image enhancement weights available on HuggingFace and applying them before sending to ChatGPT-4o is indeed a valuable suggestion. This approach could potentially improve model performance. Your suggestion provides valuable research direction for future studies. However, in our current study's revision stage, we could not implement this suggestion due to time constraints with study completion and the significant time required to apply image enhancement to our entire dataset.
Comment 6. What about to consider the latest version of ChatGPT (OpenAI's o1)?
Response 6:
There are several important reasons for using ChatGPT-4o in our current study. When our research began, ChatGPT-4o was the latest version of ChatGPT, and our study protocol, ethics committee application, and methodology were designed around this version. While our data collection and analysis process was progressing according to a predetermined timeline, OpenAI-o1 was released. At this stage, re-evaluating our entire dataset with the new version could have created methodological consistency issues and significantly extended our study completion time. Additionally, an important factor to consider was that our ethics committee approval was specifically obtained for the ChatGPT-4o version.
We will consider your suggestion in future studies.
Comment 7. It is recommended but not necessary that it's better to consider some ChatGPT comparison tools (Google Gemini:) also for this study
Response 7:
We couldn't implement this suggestion in our current study because our ethics committee application methodology specifically aimed to test ChatGPT-4o's ICH detection ability, and evaluating our dataset with other AI types would not align with this approved methodology. However, your suggestion provides an important research perspective for future studies. A comparative analysis of different AI models' (ChatGPT, Google Gemini, and others) performance in ICH detection would be valuable for understanding how technological developments translate to clinical applications.
We are grateful for the time you have dedicated to our study. Your recommendations have guided us in preparing a more enhanced and structured version of our study. We hope that our revisions meet your expectations.
Sincerely

Round 2
Reviewer 1 Report
Comments and Suggestions for Authors
Thank you for the revisions.
Author Response
Dear Reviewer,
Thank you very much for your support.
Best regards,